# Genetic Diversity of *Epichloë* Endophytes Associated with *Brachypodium* and *Calamagrostis* Host Grass Genera including Two New Species

**DOI:** 10.3390/jof8101086

**Published:** 2022-10-15

**Authors:** Adrian Leuchtmann, Christopher L. Schardl

**Affiliations:** 1Institute of Integrative Biology, ETH Zürich, CH-8092 Zürich, Switzerland; 2Department of Plant Pathology, University of Kentucky, Lexington, KY 40546-0312, USA

**Keywords:** *Brachypodium*, *Calamagrostis*, *Epichloë* endophytes, microsatellite, phylogenetic analysis, taxonomy

## Abstract

Fungi of genus *Epichloë* (Ascomycota, Clavicipitaceae) are common endophytic symbionts of Poaceae, including wild and agronomically important cool-season grass species (subfam. Poöideae). Here, we examined the genetic diversity of *Epichloë* from three European species of *Brachypodium* (*B. sylvaticum, B. pinnatum* and *B. phoenicoides*) and three species of *Calamagrostis* (*C. arundinacea*, *C. purpurea* and *C. villosa*), using DNA sequences of *tubB* and *tefA* genes. In addition, microsatellite markers were obtained from a larger set of isolates from *B. sylvaticum* sampled across Europe. Based on phylogenetic analyses the isolates from *Brachypodium* hosts were placed in three different subclades within the *Epichloë typhina* complex (ETC) but did not strictly group according to host grass species, suggesting that the host does not always select for particular endophyte genotypes. Analysis of microsatellite markers confirmed the presence of genetically distinct lineages of *Epichloë*
*sylvatica* on *B. sylvaticum*, which appeared to be tied to different modes of reproduction (sexual or asexual). Among isolates from *Calamagrostis* hosts, two subclades were detected which were placed outside ETC. These endophyte lineages are recognized as distinct species for which we propose the names *E. calamagrostidis* Leuchtm. & Schardl, sp. nov. and *E. ftanensis* Leuchtm. & A.D. Treindl, sp. nov. This study extends knowledge of the phylogeny and evolutionary diversification of *Epichloë* endophytes that are symbionts of wild *Brachypodium* and *Calamagrostis* host grasses.

## 1. Introduction

In plant parasitic fungi, host association is expected to be a major driver for the emergence of new species given their dependence on the host [1,2]. Underlying mechanisms of speciation may include cospeciating between parasite and host, host-range expansion and host jumps [3,4]. Although most well-studied cases of speciation either concern model systems or come from studies in agronomic settings, fewer examples have been investigated from natural ecosystems.

*Epichloë* species (Ascomycota, Clavicipitaceae) are common endophytic symbionts of grasses of the subfamily Poöideae [5,6]. They systemically infect above-ground parts of the host plant, and for reproduction may exhibit two different life history strategies. Sexually reproducing species form fruiting structures (called stromata) producing ascospores around developing grass inflorescences, thereby preventing host flowering and seed set (choke disease). Ascospores are released into the air and mediate contagious spread to new hosts. Most asexual species cause no symptoms in their hosts, grow into host ovules and seeds and thus are clonally propagated. [7,8]. In some grass/endophyte associations intermediate levels of choking occur, which allow for both strategies of fungal reproduction [9]. The type of reproduction is usually characteristic of a species or strain in association with a particular host and appears to be controlled mostly by the genotype of the fungus [10,11,12]. However, in some species environmental factors such as high soil nutrient level following fertilization can reduce or prevent disease expression [13,14].

The genus *Epichloë* currently encompasses 39 species, of which 13 are stroma-forming and mostly sexual, and 26 are asexual [6,15,16,17,18,19]. Furthermore, among the asexual species the majority are allopolyploid, interspecific hybrids that presumably resulted from hybridizations between two or more ancestral sexual species [20]. Most sexual *Epichloë* species were originally identified as distinct mating populations (MP) based on their interfertility, and a biological species concept has generally been applied in their descriptions [21,22,23,24,25]. The largest MP is represented by several taxa of the *Epichloë typhina* complex (ETC), which includes *E. typhina* (Pres.) Brockm. With ssp. *Clarkii* and spp. *Poae*, and *E. sylvatica* Leuchtm. & Schardl with ssp. *Pollinensis*, which together may infect at least 15 different host grass species [6]. More recent investigations, however, suggest that subspecies status of *E. clarkii* J.F. White and *E. poae* Tadych, K.V. Ambrose, F.C. Belanger & J.F. White may no longer be tenable, and these taxa should be recognized as species [25,26].

In recent years, numerous investigations have been made on the diversity of *Epichloë* species symbiotic with particular grass hosts. These hosts included agronomically important species of genus *Festuca* and *Lolium* [27,28,29], as well as genera of grasses growing naturally in woodlands and prairies [16,30,31,32,33]. A well-studied host of natural habitats in woodlands is the grass species *Brachypodium sylvaticum* (Huds.) Beauv. (false brome), which is widely distributed in temperate Eurasia and North Africa [34,35] and appears to be almost ubiquitously infected by *Epichloë sylvatica* Leuchtm. & Schardl in its native Eurasian range [10,36,37,38,39]. *Epichloë sylvatica* is mainly associated with *B. sylvaticum*, but may rarely be found on *Hordelymus europaeus* (L.) Harz [21,31]. The endophyte can reproduce sexually, which involves outcrossing between individuals with different mating types within a distinct MP. However, expression of stromata is rarely observed in infected populations of *B. sylvaticum*, but when it is, either stroma formation is restricted to a subset of infected plants, or individuals have both choked and healthy flowering tillers [10]. Moreover, plants with mixed symptoms are typically infected by more than one endophyte genotype, which may be responsible for the different disease expression [12,40].

Two other species of *Brachypodium*, *B. pinnatum* (L.) B. Beauv. and *B. phoenicoides* (L.) Roem. & Schult., are occasionally found to be infected with stroma-forming *Epichloë* [21,39,41]. Isolates from these hosts have been assigned to *E. typhina* based on their mating compatibility and genetic relatedness. However, isolates infecting *B. pinnatum* are not uniform and some may have a closer phylogenetic relationship to *E. sylvatica* than to *E. typhina* [6].

A second host genus of *Epichloë* examined in the present study is *Calamagrostis* (reed grass) which is distributed around the globe in temperate zones and comprises approximately 250 species that are particularly rich in South America [42]. However, a minority of the species have been examined so far and only approximately 18 species are reported to be infected. These included host species with symptomless infections mainly in South America [43,44] and species that had stromata. A stroma-forming *Epichloë* species, *E. amarillans* J.F. White, was identified from North American *C. canadensis* (Michx.) P. Beauv. [45], whereas collections from European *Calamagrostis* species have been assigned to *E. baconii* J.F. White [46]. A third stroma-forming species, *E. stromatolonga* (Y.L. Ji, L.H. Zhan & Z.W. Wang) Leuchtm., was more recently described from *C. epigejos* (L.) Roth in China [47].

In this study we report on the genetic diversity of *Epichloë* species symbiotic with *Brachypodium* and *Calamagrostis* species in Europe. Particular emphasis is given to endophytes infecting *B. sylvaticum*, which were obtained from a large sample across Europe. To evaluate diversity we use microsatellite markers and DNA sequence data from *tubB* and *tefA* genes. Two endophyte lineages infecting *Calamagrostis* hosts are recognized as new species. This study extends our knowledge on the phylogeny and evolutionary diversification of *Epichloë* endophytes in wild *Brachypodium* and *Calamagrostis* host grass species.

## 2. Materials and Methods

### 2.1. Host Grass Species

The genus *Brachypodium* has approximately 20 species classified in its own tribe Brachypodieae [48]. Among these, three species have been found to be infected by *Epichloë* endophytes. (1) *Brachypodium sylvaticum* is a perennial, caespitose (tussock forming) grass that is native to Europe, temperate Asia and north-western Africa, and introduced to other parts of the world, e.g., Pacific North West of the United States [49]. It is most commonly found in forests and woodlands, preferring the moist, shaded canopy, but may also grow in open areas. In contrast to the other species of the genus that are predominantly wind pollinated, a very high level of selfing is observed in *B. sylvaticum* [35]. This host is commonly infected by *E. sylvatica* [21]. (2) *Brachypodium pinnatum* is a perennial, stolon-forming grass with similar distribution as *B. sylvaticum* in temperate Eurasia. It prefers dry, nutrient poor grassland or light deciduous woods and may be a pioneer in open soil or clearings. It can be infected by choke-forming *E. typhina* which is not seed-transmitted on this host [6]. (3) *Brachypodium phoenicoides* is also stolon-forming and grows in light oak woods in the south-western part of the Mediterranean region. This species may be infected by stroma-forming *E. typhina* [41].

The genus *Calamagrostis* of tribe Agrostideae comprises approximately 250 species worldwide. In our study, we examined endophytes from four *Calamagrostis* hosts. (1) *C. villosa* (Chaix) J.F. Gmel., a stolon-forming species occurring on acid soils in the understory of pine and larch forest in central and southern European mountains, (2) *C. varia* (Schrad.) Host that grows on lime rich soils preferably in open scree or other pioneer sites and has a similar distribution, and (3) *C. purpurea* (Trin.) Trin. that grows on humid to damp sites in woodlands or aside standing water with a native range from subalpine to subarctic. *C. purpurea* has been repeatedly reported to be infected by stroma-forming *Epichloë* mainly in Nordic European countries but without assigning a distinct species name to it [36,50]. (4) *C. arundinacea* (L.) Roth has a Euro-Siberian distribution and occurs on mostly lime-free, dry to humid soils, preferentially in mountain woodlands.

### 2.2. Endophyte Sampling and Isolation

Sampling was conducted in different years from host grasses growing at natural sites. Grass shoots (with or without stromata) or seeds were collected and taken to the laboratory. One isolate from *B. sylvaticum* was obtained from Japan as a cultured strain. Geographic origin, source of isolation and mode of reproduction of all isolates used in this study are listed in Table 1. In addition, location of collection sites of *Epichloë*-infected *Brachypodium* hosts are indicated in Figure 1.

Endophytes were isolated from surface-disinfected plant tissues (culm) or from seeds that were placed on standard nutrient agar medium supplemented with 50 mg/L oxytetracycline (Pfizer, New York, NY, USA) in Petri dishes as previously described [51]. Isolates from stroma-forming plants were either obtained from ascospores or in some cases from mycelium taken from the innermost part of young, clean stromata after splitting them open under sterile condition. Colonies growing from plant tissues, seeds or mycelium were checked under the microscope for purity, and *Epichloë* identity was confirmed based on the characteristic sporulation [52]. For DNA extraction isolates were grown on V-8 liquid medium on a rotary shaker for 12–14 days as previously described [31].

### 2.3. Microsatellite Analyses

Genomic DNA was extracted from freeze-dried mycelia with the NucleoSpin Plant II Kit (Macherey-Nagel, Düren, Germany) following the manufacturer’s instructions.

Microsatellite analyses followed the protocol developed by Schirrmann et al. [26] using a multiplex PCR approach. The 16 markers were arranged in four multiplex sets (Table 2) and each set was amplified using approximately 1 ng of genomic DNA in a PCR volume of 10 µL. Amplification conditions were as follows: initial denaturation of dsDNA at 94 °C for 3 min, 30 cycles of 30 s denaturation at 94 °C with 1 min annealing at 56 °C and 30 s extension at 72 °C, and final elongation at 72 °C for 5 min. Signals of the PCR products were detected on a 3130xl DNA Analyzer (Applied Biosystems, Foster City, CA, USA) with GeneScan-500 LIZ as size standard. Electropherograms were analyzed using Geneious 9.1.8 (Biomatters, Auckland, New Zealand).

Principal coordinate analysis (PCoA) was applied to microsatellite data to visualize proximity of isolates based on genetic distances using GenAlEx 6.5 [53].

### 2.4. DNA Sequencing and Phylogenetic Analyses

PCR amplification and sequencing was performed as previously described [31]. Amplification of the translation elongation factor 1-alpha gene (*tefA*) segment including introns 1–4 employed primer pair 5′-GGG TAA GGA CGA AAA GAC TCA-3′ and 5′-CGG CAG CGA TAA TCA GGA TAG-3′. Amplifications of the β-tubulin gene (*tubB*) segment including introns 1–3 employed primer pair 5′-TGG TCA ACC AGC TCA GCA CC-3′ and 5′-TGG TCA ACC AGC TCA GCA CC-3′ [46]. Reactions were performed in a final volume of 25 µL containing about 10 ng of genomic DNA in a standard touchdown cycle. Labelling reactions involved BigDye^®^ Terminator v3.1 chemistry (Applied Biosystems™, Foster City, CA, USA) and sequences were obtained on a 3130xl Genetic Analyzer (Applied Biosystems™, Foster City, CA, USA). New sequences of this study have been submitted to GenBank (www.ncbi.nlm.nih.gov) and their accession numbers are indicated in bold in Table 1.

After manually editing, sequences were aligned with MUSCLE implemented in Geneious 9.1.8 (Biomatters, Auckland, New Zealand) including previously published sequences and selected reference sequences. Phylogenetic trees were inferred by maximum likelihood (ML) with likelihood settings of best-fit models selected by automated AICc model selection using PAUP* 4.0a [54]. The ML trees were generated in a heuristic search with gaps treated as missing information and random sequence additions. Bootstrap support values were estimated from 100 ML replications with random number seed and stepwise sequence addition.

### 2.5. Morphological Examinations

Colony growth was examined from cultures on potato dextrose agar (PDA; BD Comp., Sparks, MA, USA). Petri dishes were inoculated with 2 mm agar blocks, sealed and incubated at 24 °C in the dark. Colony diameter was measured after 21 days from three replicates per strain, and cultures were characterized and photographed. Microscopic observations of asci and ascospores, conidiogenous cells and conidia were made with an Olympus BX40 microscope (Olympus Corp., Tokyo, Japan) from stromata of herbarium specimens and from mycelium grown on PDA mounted in lactic acid. Measurements were taken with an ocular micrometer at 400× or 1000× using phase contrast optics. Of each fungal structure 20 measurements were taken and range with average in parenthesis is given. All specimens examined are deposited at the Fungarium of ETH Zürich (ZT).

## 3. Results

### 3.1. Microsatellite Data

Microsatellite data were obtained from 30 endophyte isolates from *B. sylvaticum*, 11 isolates from *B. pinnatum* and one isolate from *B. phoenicoides* (Table 1, Appendix A). Principal coordinate analysis (PCoA) of haplotypes based on 16 microsatellite loci grouped isolates in three main clusters (Figure 2). One cluster contained all but one asexual isolate from *B. sylvaticum* (Bsa), a second cluster included the majority of the sexual isolates from *B. sylvaticum* (Bss) and one asexual isolate (Bs 9301), and a third cluster included the remaining Bss isolates and isolates from *B. pinnatum* (Bp) and *B. phoenicoides* (Bph). In this cluster, Bss and Bp isolates tended to be separated from each other forming two subclusters. A separate PCoA made with only asexual isolates from *B. sylvaticum* revealed some subclustering, but clusters did not reflect geographic origin (Appendix A). Likewise, PCoA of isolates from *B. pinnatum* showed subclusters that in part were related to the site of origin (population) but not to the overall geography (Appendix A).

### 3.2. *tubB* Phylogeny

Isolates from *B. sylvaticum* (Bs), *B. pinnatum* (Bp) and *B. phoenicoides* (Bph) grouped within the *E. typhina* complex but did not form separate subclades according to different host plant species (Figure 3). For example, subclade I and IV of the *tubB* phylogeny included both Bp and Bs isolates. Subclade I comprised most Bs isolates together with two Bp isolates from Perroudaz, Switzerland and Basilicata, Italy. Subclade IV was placed at the base of the *E. typhina* main clade next to the isolates of *E. poae* and included the remaining Bp isolates together with a Bs isolate from Salamanca, Spain, and *E. sylvatica* ssp. *pollinensis*, a taxon that has been described from *Hordelymus europaeus* in Southern Italy. The isolate from *B. phoenicoides* (Bph) grouped in subclade III with *E. poae*, but without significant bootstrap support.

The *tubB* phylogeny placed the new isolates from *Calamagrostis* species in subclades of a main clade that encompassed five haploid *Epichloë* species described to date. The three isolates from *C. arundinacea* (Ca) had identical *tubB* sequences, which grouped them in a well-supported subclade with sequence relationship to *E. stromatolonga*, an Asian species described from *Calamagrostis epigejos*. The isolate from *C. purpurea* (Cp) and three isolates from *C. villosa* (Cv) grouped in a well-supported subclade that formed a polytomy with the *E. baconii* subclade and another subclade that included *E. festucae* Leuchtm., Schardl & M.R. Siegel, *E. mollis* (Morgan-Jones & W. Gams) Leuchtm. & Schardl, *E. stromatolonga*, and the Ca isolates.

### 3.3. *tefA* Phylogeny

In the *tefA* phylogeny all isolates from *Brachypodium* hosts grouped in a moderately supported clade in the *E. typhina* complex (Figure 4). As in the *tubB* tree, Bs and Bp isolates were not resolved by host species in the *tefA* tree. A moderately supported subclade (Ia) of the *tefA* tree encompassed all asexual Bs isolates together with some of the sexual Bs isolates. Otherwise, the clade exhibited no resolution based on host or geography.

The *tefA* phylogeny indicated the same relationships of isolates from *Calamagrostis* as did the *tubB* phylogeny. Considering only well-supported branches, the phylogenetic relationships to other members of that clade were consistent with the *tubB* phylogeny. One strongly supported clade included the three Ca isolates, and another strongly supported clade included the one Cp and three Cv isolates.

## 4. Discussion

Evolutionary diversification of parasites often depends on the association of infecting organisms with a particular host. Here, we examined the *Epichloë* endophytes found on different host grass species of genus *Brachypodium* and *Calamagrostis*. Following the concept of previously defined mating populations that were based on interfertility tests, *B. sylvaticum* is the host of a distinct species recognized as *E. sylvatica,* whereas *B. pinnatum* and *B. phoenicoides* is infected by the wide-host-range species *E. typhina* [21,39,41]. Microsatellite data obtained in the current study suggest that *Brachypodium* infecting *Epichloë* endophytes are closely related but are differentiated into four distinct clusters. Isolates from *B. sylvaticum,* referred here to *E. sylvatica,* form three clusters that differ in their mode of reproduction (one asexual, two sexual), while isolates from *B. pinnatum* and *B. phoenicoides* form a fourth cluster most closely related to one of the sexual *E. sylvatica* clusters (Figure 2). As inferred from DNA sequence analysis, the European isolates from *Calamagrostis* grouped into distinct clades within a larger clade that encompassed five previously described *Epichloë* species. Based on these relationships, we propose the new species *E. calamagrostidis* associated with *C. purpurea* and *C. villosa*, and *Epichloë ftanensis* associated with *C. arundinacea*.

A remarkable finding of this study was that asexual isolates from *B. sylvaticum* showed very little variation across distant locations in Europe, which spanned over several thousand kilometers (ca. 2500 km) and represented most of the latitudinal distribution range of the host. This is in stark contrast to another woodland grass, *H. europaeus*, that was found to be infected by six different endophyte taxa along a longitudinal transect across Europe (Oberhofer and Leuchtmann 2012). This finding may be explained by the pattern of recolonization of land masses after the last glaciation [55] involving infected plants of *B. sylvaticum* that contained only one endophyte lineage. Rapid spread of *B. sylvaticum* plants with its seed transmitted endophyte is promoted by the awned seeds of the species which can easily attach to the fur of animals. Sexual isolates of *E. sylvatica* were genetically more variable and were often distinct from the asexual isolates, suggesting that for the sexual lineages either multiple colonization events have occurred or infections have been acquired later from long distance dispersal of ascospores. The genetic distinctness of sexual and asexual isolates of *E. sylvatica* has been previously reported in populations from Switzerland based on allozyme data [10]. The current study with a European wide sample making use of microsatellite data confirms this finding.

Sequence data of *tubB* and *tefA* genes were not concordant with microsatellite data and did not always resolve *E. sylvatica* isolates with different modes of reproduction in separate clades. Sexual and asexual isolates were included in the same clade or even had identical sequences, although some substructuring was evident. This may suggest that genetic differentiation of the asexual, clonally propagated isolates occurred relatively recently and that sexual and asexual lineages share a common origin.

Without exception all populations of *B. sylvaticum* that have been examined so far in Europe were endophyte infected, usually with an incidence of 100%. In light of the expectation that seed transmission is not perfect and at least a low percentage of uninfected seeds is produced [56]. This observation suggests that the symbiosis of this host with seed-transmitted *E. sylvatica* provides a selective advantage. However, it is still unclear what the benefits for the host of endophyte infection are, unlike in many other endophyte associations. For example, *E. festucae* and *E. coenophiala* (Morgan-Jones & W. Gams) C.W. Bacon & Schardl are known to produce a variety of alkaloids with distinct antiherbivore properties [57,58]. Previous analysis of *E. sylvatica*-infected *B. sylvaticum* plants indicated an absence of peramine (a pyrrolopyrazine), known lolines (aminopyrrolizidines) and ergopeptines (ergot alkaloids) [59]. Genome sequence analysis indicated no functional genes for ergot alkaloids, aminopyrrolizidines or indole-diterpenes, and a *ppzA* allele lacking the R domain for peramine (C.L. Schardl, unpublished data). However, ppzA genes likely determine biosynthesis of related pyrrolopyrazines, as recently described [60]. In an in vitro study using the model herbivore *Spodoptera frugiperda,* uninfected *B. sylvaticum* leaves were preferred over infected leaves suggesting that increased resistance to herbivores based on unknown factors could indeed play a role [61]. Such factors may include one or more pyrrolopyrazines that are products of the *ppzA* gene, compounds determined by other secondary metabolite gene clusters [62] or even an anti-insect protein such as is encoded by the *mcf*-like gene [63].

Additional endophyte benefits that could be involved and are documented for other endophyte-host systems include improved seedling establishment and growth promotion, which can improve competitive abilities of infected plants [64,65]. However, experiments testing competitive abilities of *B. sylvaticum* plants did not confirm this hypothesis, but rather endophyte infection had a negative effect on plant growth [66].

Another potential benefit of the endophyte is enhanced seed production or survival. It is worth noting that the *Epichloë* species (including *E. sylvatica*) apparently possess the *fhb7* gene that was acquired by a wheat wild relative and encodes an enzyme for detoxification of trichothecenes which are virulence factors of the wheat head blight pathogen, *Fusarium graminearum* [67]. This raises the possibility that the endophyte may provide protection from such a pathogen, which prevents reduced seed yield.

Sequence data were used to infer the phylogenetic position of *Brachypodium* endophytes within the *E. typhina* complex. In the *tubB* phylogeny isolates from these hosts were distributed among two subclades, subclade *sylvatica* (I) and subclade *pollinensis* (IV) confirming previous findings [6]. The exception was placement of the isolate from *B. phoenicoides* in subclade *poae* (III), but without significant bootstrap support. Inspection of the alignment (Appendix A) suggested a possible intragenic recombination within *tubB* occurred between *Brachypodium-* and *Poa*-associated lineages. In the *tefA* tree, however, all isolates from *Brachypodium* hosts were in the same clade (I) and the subclade that included ssp. *pollinensis* was nested within that clade, suggesting that isolates from *Brachypodium* hosts share the same gene pool. The apparent paraphyly of *Brachypodium*-associated *Epichloë* in the *tubB* tree may suggest complicated origins of the *Epichloë* lineages occurring in symbiosis with *Brachypodium* species, or it may be a lineage sorting effect. Moreover, isolates from *B. sylvaticum* and *B. pinnatum* were not strictly grouped in separate subclades, indicating that endophytes of the two hosts may not be host specific, but rather can move between hosts. For example, sequences of an isolate from *B. sylvaticum* (8928/2) were identical with two isolates from *B. pinnatum* (9640/1, 9435/1) in the *tefA* phylogeny, but not in the *tubB* phylogeny.

Isolates found on *Calamagrostis* hosts formed two distinct clades in the *Epichloë* phylogeny, part of a larger clade that included *E. amarillans*, *E. baconii*, *E. festucae* and *E. mollis,* and one next to *E. stromatolonga* (Figure 3 and Figure 4). *Epichloë stromatolonga* is an asexual, stroma-forming species that has been described on *C. epigejos* from a single location in China [47]. Our isolates from *C. arundinacea* (Ca) were relatively close to this species based on *tubB* and *tefA* sequences and thus placed in the same subclade with 100% or 90% bootstrap support, respectively. However, given the distinct host relation and occurrence on different continents, we suggest that this endophyte represents an independent phylogenetic lineage that may have evolved in Europe. This endophyte should therefore be recognized as a distinct species, for which we propose the new name *E. ftanensis*. Mating tests have not been performed, but phylogenetic distance to other mating populations (MP) suggest that the new species (perhaps together with *E. stromatolonga*) represents a distinct MP. The new species has so far been found only at one site on *C. arundinacea* located in a remote valley of the Central Alps of Switzerland. More intensive sampling of *C. arundinacea* in a wider area would be necessary to assess its actual distribution.

The second clade with *Calamagrostis*-derived isolates contained isolates from *C. villosa* and *C. purpurea.* Isolates from *C. villosa* have been previously assigned to *E. baconii* based on their sexual compatibility with this species that otherwise infects *Agrostis* spp. [21]. Incidentally, isolates from *C. villosa* were genetically distinct both in allozyme patterns and sequences of three nuclear genes *act1*, *tubB* and *tefA* [46]. Our phylogenetic analyses of additional isolates from the same and from a second *Calamagrostis* host placed these isolates in a well resolved clade distinct from *E. baconii* (Figure 1 and Figure 2). The observed interfertility between *E. baconii* and *Calamagrostis*-derived isolates in mating tests poses a conflict between biological and phylogenetic species concepts. However, successful outcrossing observed experimentally does not necessarily mean that fertile mating between species occurs in nature. *Agrostis* and *Calamagrostis* hosts occupy different niches in grass- or woodlands and may not grow close enough for mutual gamete transfer by *Botanophila* flies. If rare mating still would occur, progeny may be lost due to lack of a compatible host because the two species are expected to be host adapted, as has been observed in other interfertile *Epichloë* species [68]. Strains from *C. purpurea* have not been tested and it is not known whether they are interfertile with *E. baconii*. The genetic distinctness of isolates from the *Calamagrostis* hosts as shown previously and in the current study suggests that there is no gene flow between *E. baconii* and *E. calamagrostidis* and that the *Calamagrostis*-associated strains represent a reproductively isolated phylogenetic species, despite not yet fully establishing sexual barriers.

## 5. Taxonomy

***Epichloe calamagrostidis*** Leuchtm. & Schardl, sp. nov., Figure 5

MycoBank MB846049

*Colonies* on PDA white to tan, in the center cottony and raised, towards the margin flattened with sparse aerial mycelium, margin even or slightly wavy, reverse of colony distinctly brownish, moderate growing, attaining 23–26 mm diameter in 21 days at 24 °C. Sporulation in culture sparse to moderate. Conidiogenous cells phialidic, arising perpendicularly from hyphae, hyaline, 10–22 µm long, 1–2 µm wide at base and gradually at-tenuating to approximately 0.5 µm at the tip, with or without a basal septum. Conidia ellipsoidal to reniform, hyaline, aseptate, 3.3–4.9 (3.9) × 1.6–2.5 (2.0) µm. *Stromata* on flowering culms enclosing undeveloped inflorescence, leaves and sheath of flag leaf, cylindrical, 18–35 mm long and 2–3 mm wide, in early stage white and covered with a dense layer of conidiogenous cells producing conidia 4.0–5.6 × 2.0–3.0 µm in size. Upon fertilization stromata become orange to brownish with perithecia forming on top of the conidial stroma. Perithecia pyriform, approximately 140–160 µm wide, 240–280 µm high, light orange, 40–45 per mm^2^. Asci cylindrical with a short tapering stalk, bearing a distinct hemispherical apical cap with central pore, 200–280 × 4.8–7.2 µm, containing eight ascospores. Ascospores hyaline, filiform, 225–290 × 1.5–1.7 µm, non-septate within ascus, becoming multiseptated at full maturity, not disarticulating. Endophyte in infected flowering tillers not seed-transmitted.

**Figure 5 jof-08-01086-f005:**
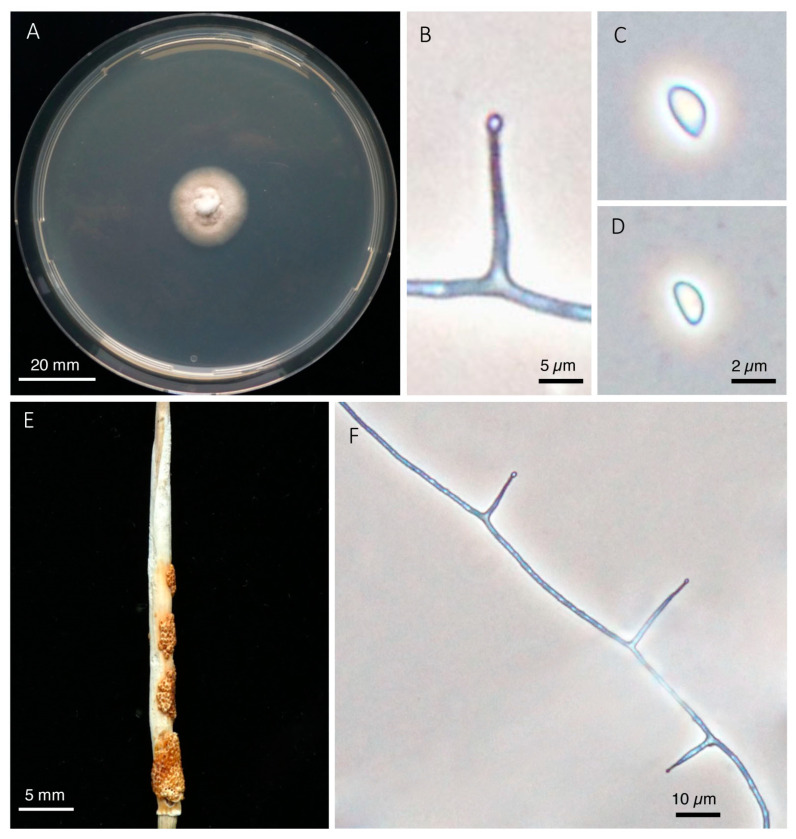
*Epichloë calamagrostidis*. (**A**). Colony grown on PDA for 21 days at 24 °C. (**B**). Conidiogenous cell with emerging conidium. (**C**,**D**). Conidia. (**E**). Stroma with mature perithecia on *C. varia* (coll. H. Seitter). (**F**). Sporulating hypha in culture.

*Etymology*: Referring to the Host Genus *Calamagrostis*.

*Holotype*: SWITZERLAND, Ct. Grisons, Lavin, stromata on *Calamagrostis villosa*, 8 September 1990, leg. A. Leuchtmann (ZT Myc 99902); ex type cultures AL9039, ATCC 200745; GenBank accession nos. L78270 (*tubB*), AF231196 (*tefA*).

*Additional specimens examined*: SWITZERLAND, Ct. Grisons, Lavin, stromata on *Calamagrostis villosa*, 8 September 1990, leg. A. Leuchtmann, culture AL9040; Ct. Grisons, Bever, stromata on *Calamagrostis villosa*, 24 July 1996, 5 August 2022, leg. A. Leuchtmann, cultures AL9618, AL2016, CBS 147678; Ct. Grisons, Bregaglia, Roticcio, stromata on *Calamagrostis villosa,* 7 July 1997, leg. G. Meijer; Ct. Grisons, Bregaglia, Bondo, stromata on *Calamagrostis villosa,* 4 August 2022, leg. A. Leuchtmann; Ct. Obwalden, Lungern, Schönbüel, stromata on *Calamagrostis varia,* 2 August 1991, leg. A. Leuchtmann; Ct. Bern, Hasliberg, Wasserwendi, stromata on *Calamagrostis varia*, 6 August 1918, leg A. Thellung. LICHTENSTEIN: Triesen, Tuasswand, stromata on *Calamagrostis varia*, 6 June 1974, leg. H. Seitter. FINLAND, Oulu, Paltamo, stromata on *Calamagrostis purpurea*, July 2007, leg. P. Wäli, culture AL0908; Paltamo, Melalahti, on *Calamagrostis purpurea*, 30 August 1960, leg. T. Ulvinen.

*Hosts: Calamagrostis purpurea, C. varia, C. villosa*.

*Known distribution*: Switzerland, Finland.

*Comments*: Besides its genetic distinctness this species is characterized by relatively short stromata no longer than 35 mm, and short ascospores that do not disarticulate. Ascospores of the related *E. baconii* break into several part-spores while still within the ascus. We therefore consider *Epichloë* isolates from *C. villosa* and *C. purpurea*, and most likely also from *C. varia* that could not be sequenced, as distinct species and propose the name *E. calamagrostidis*, sp. nov. *Epichloë calamagrostidis* can form symptomless infections in *Calamagrostis* spp., because host grasses often do not flower (AL, personal observation). However, if infected plants occasionally form inflorescences instead of stromata the endophyte does not invade seeds, thus qualifying this association as type I [9].

An interesting observation was that stromata in populations of *C. villosa* and *C. varia* often remained unfertilized or, if fertilized, the perithecia were barren. Similar observations have been made for *E. brachyelytri* Schardl & Leuchtm. that rarely formed perithecia with ascospores [22], and in two other species, *E. stromatolonga* and *E. scottii* T. Thünen, Y. Becker, M.P. Cox & S. Ashrafi, fertilized stromata have never been observed [18,47]. A possible explanation for these observations is that endophytes at a particular site exist as clones that represent only one mating type, which prevents successful mating. Barren perithecia can be the result of interspecific matings with spermatia from other *Epichloë* species that may have been present in the area. Host grasses of *C. villosa* and *C. varia* do in fact propagate vegetatively through subterranean stolons resulting in large clonal patches. Alternatively, there may be no *Botanophila* flies present at these sites restricting efficient gamete transfer [69]. However, at least at the Bondo site where fertilized stromata on *C. villosa* occurred, signs of fly visitation (egg shells and feeding damage) on the stromata were observed suggesting that the first mentioned hypothesis is more likely.

***Epichloe ftanensis*** Leuchtm. & A.D. Treindl, sp. nov., Figure 6

MycoBank MB846050

*Colonies* on PDA pure white, cottony, moderately raised, margin even or slightly frayed, reverse of colony light brown, moderate growing, attaining 26–32 mm diameter in 21 day at 24 °C. Sporulation in culture very sparse. Conidiogenous cells phialidic, arising perpendicularly from hyphae, hyaline, 16–42 µm long, 1.6–2 µm wide at base and gradually attenuating to approximately 0.5 µm at the tip, basal septum lacking. Conidia ellipsoidal to reniform, hyaline, aseptate, 3.3–4.8 (4.0) × 1.9–2.6 (2.3) µm. *Stromata* on flowering tillers enclosing undeveloped inflorescence, leaves and sheath of flag leaf, cylindrical, 19–45 mm long and 2.5–3 mm wide, in early stage white and covered with a dense layer of conidiogenous cells producing conidia 4.0–5.6 × 2.0–0–3.6 µm in size. Fertilized, mature stromata turn orange, forming perithecia on top of the conidial stroma, 12–16 per mm^2^. Individual perithecia pyriform, approximately 220 µm wide, 440 µm high, wall and neck light orange, embedded in a whitish tissue of globular cells. Asci cylindrical with a short tapering stalk, bearing a distinct hemispherical apical cap that may be flattened at maturity, 250–475 × 5.5–7 µm, containing eight ascospores. Ascospores hyaline, filiform 300–460 (380) × 1.5–1.6 (1.5) µm, with up to 20 septa at maturity, not disarticulating. Endophyte in infected flowering tillers not seed-transmitted.

*Etymology*: Referring to the Collection Site Near Village Ftan.

*Holotype*: SWITZERLAND, Ct. Grisons, Ftan, stromata on *Calamagrostis arundinacea*, 23 August 2020, leg. A. Leuchtmann & A.D. Treindl (ZT Myc 66903); ex type cultures AL2015, CBS 147676; GenBank accession nos. MW283354 (*tubB*), MW283391 (*tefA*).

*Additional specimens examined*: SWITZERLAND, Ct. Grisons, Ftan, stromata on *Calamagrostis arundinacea*, 5 September 2016, leg. A. Leuchtmann, cultures AL1614/1, AL1614/2.

*Host*: *Calamagrostis arundinacea.*

*Known distribution*: Switzerland.

**Figure 6 jof-08-01086-f006:**
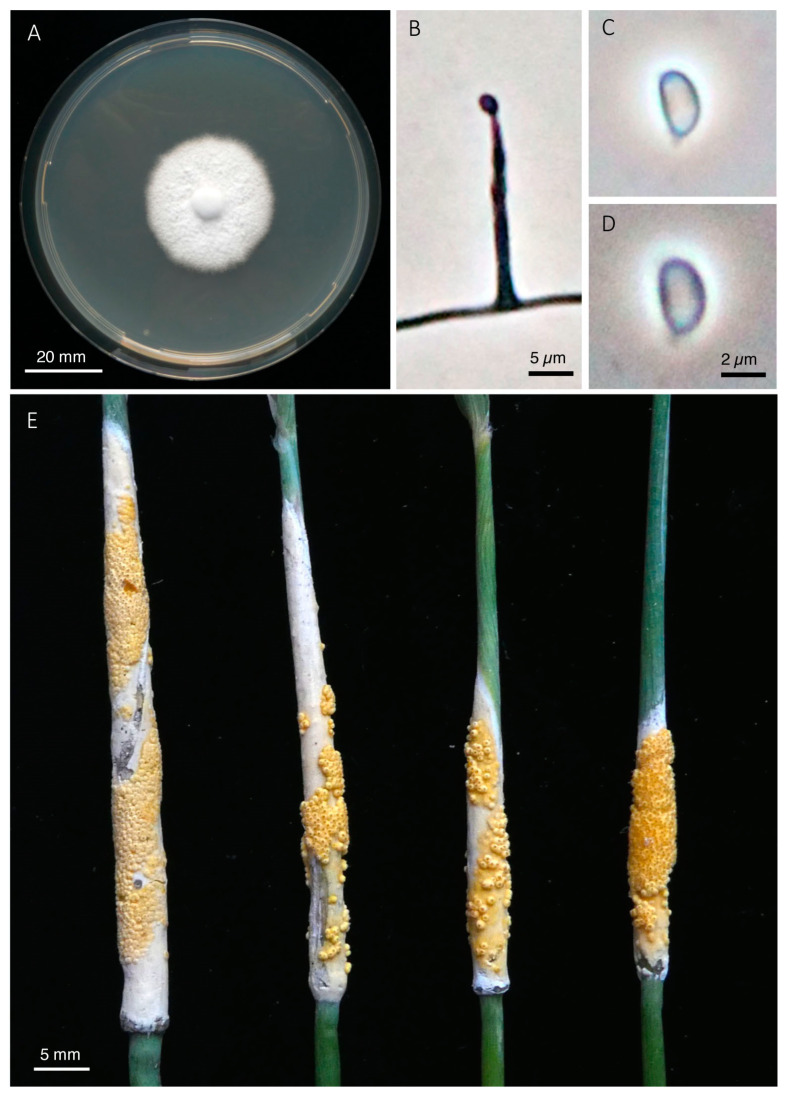
*Epichloë ftanensis*. (**A**) Colony grown on PDA for 21 days at 24 °C. (**B**) Conidiogenous cell with emerging conidium. (**C**–**D**) Conidia. (**E**) Stromata with mature perithecia (holotype).

*Comments*: Distinctive morphological features of this taxon are the very long ascospores of up to 460 µm in length, and the size of the perithecia that are larger than in most other species. Only in *E. elymi*, a species native to North America, ascospores may occasionally be longer. Fertilization of stromata at the observed site was variable with perithecia covering 0% to 80% of the stroma surface. About half of the stromata showed signs of *Botanophila* fly visitations (egg shells or larvae) suggesting that flies take an active role in fertilization and outcrossing of this taxon [70]. Our isolates are genetically close to the earlier described species *E. stromatolonga* based on *tubA* and *tefB* sequences. However, collections of *E. stromatolonga* were reported to have stromata that always remain unfertilized and lack perithecia, thus the species is effectively asexual [47]. Furthermore, this taxon differs by its exceptionally long stromata (up 186 mm), by infecting a different host grass (*C. epigejos*) and by its geographic origin in China. Therefore, we treat our sexually reproducing specimens from *C. arundinacea* as distinct species and propose the name *E. ftanensis*, sp. nov.

## Figures and Tables

**Figure 1 jof-08-01086-f001:**
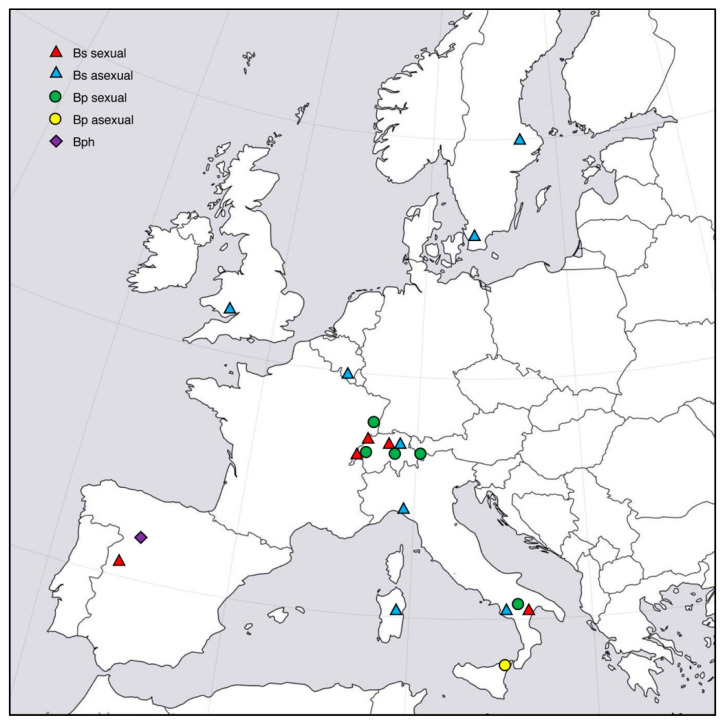
Location of collection sites in Europe of *Epichloë*-infected *Brachypodium* species: *B. sylvaticum* (Bs), *B. pinnatum* (Bp) and *B. phoenicoides* (Bph). Symbols distinguish between host, and stroma-forming (sexual) or symptomless (asexual) infections.

**Figure 2 jof-08-01086-f002:**
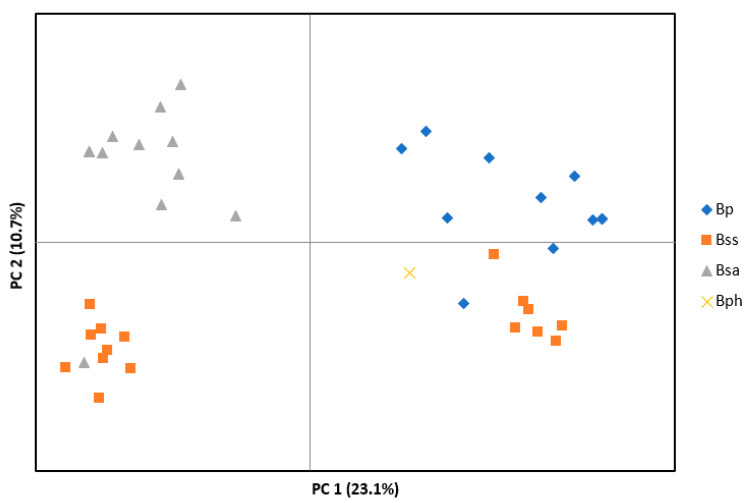
Principal coordinate analysis (PCoA) based on 16 microsatellite loci of *Epichloë* isolates from *Brachypodium* hosts: *B. sylvaticum* (Bss, sexual isolates; Bsa, asexual isolates), *B. pinnatum* (Bp) and *B. phoenicoides* (Bph).

**Figure 3 jof-08-01086-f003:**
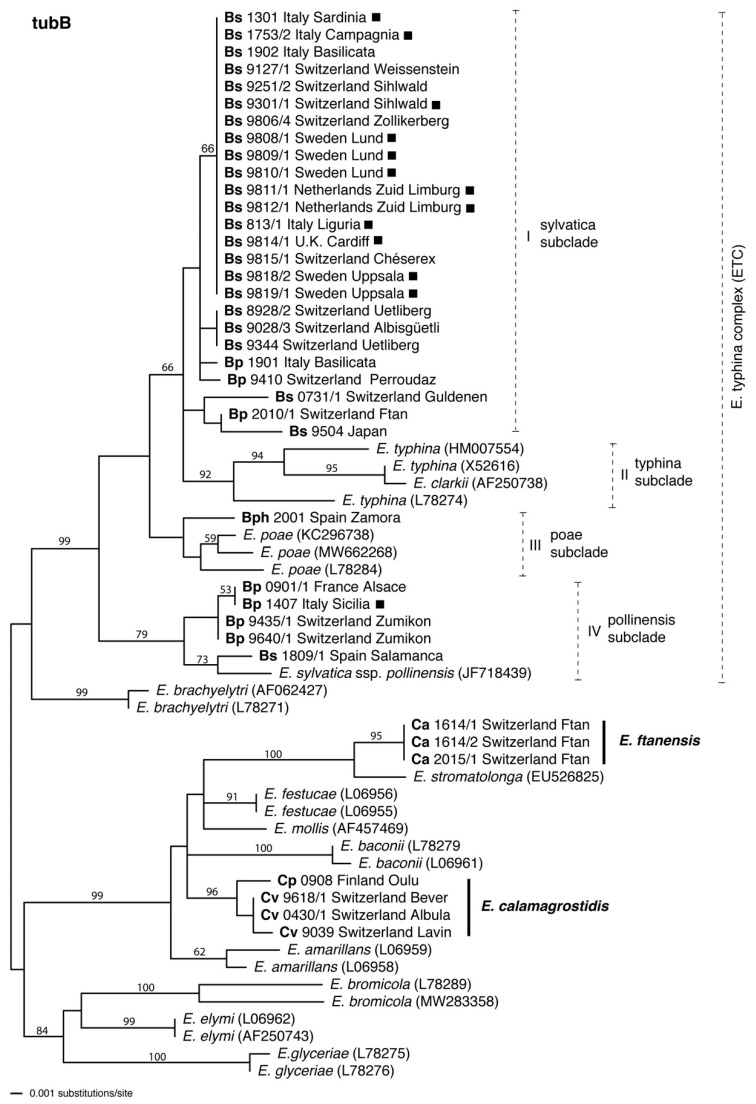
Phylogenetic tree inferred from maximum likelihood (ML) analysis of partial *tubB* gene sequences including introns 1–3 of *Epichloë* isolates obtained from *Brachypodium* and *Calamagrostis* host grasses: *B. sylvaticum* (Bs), *B. pinnatum* (Bp), *B. phoenicoides* (Bph), *C. arundinacea* (Ca), *C. purpurea* (Cp) and *C. villosa* (Cv). Asexual isolates are indicated with filled squares. representative sequences of all other sexual *Epichloë* species or subspecies described to date are included. Dashed lines with Roman numerals denote distinct subclades. The tree is midpoint rooted at the left edge. Branch support values were estimated by 100 ML bootstrap replicates and are indicated if above 50%.

**Figure 4 jof-08-01086-f004:**
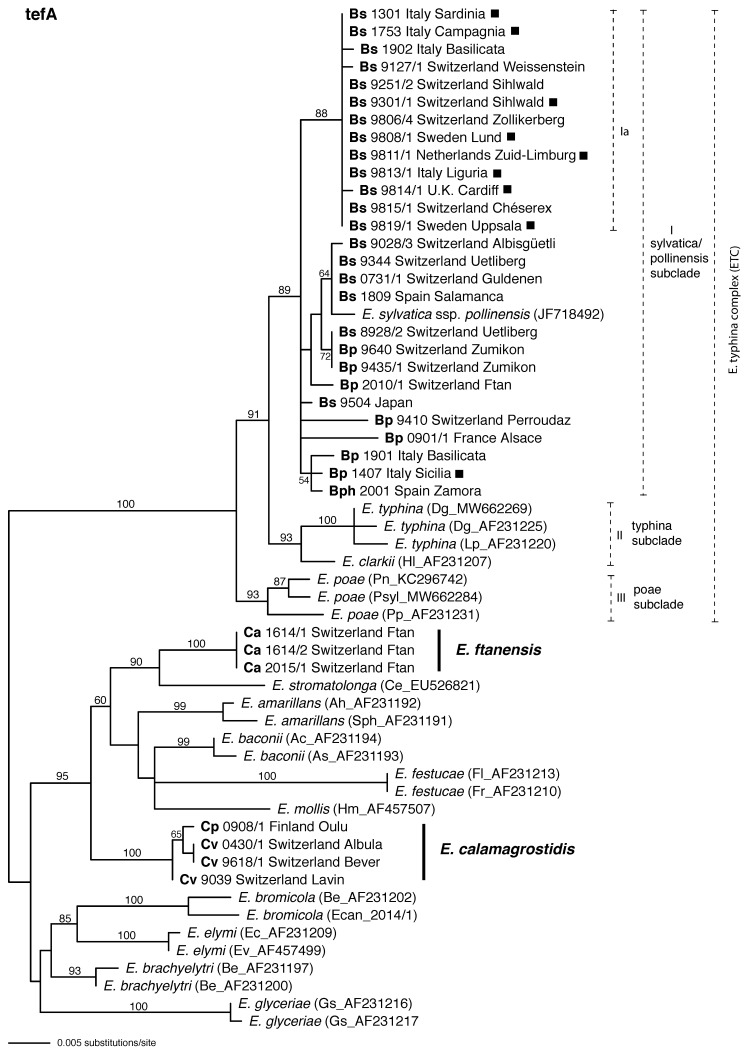
Phylogenetic tree inferred from maximum likelihood (ML) analysis of partial *tefA* gene sequences including introns 1–4 of *Epichloë* isolates obtained from *Brachypodium* and *Calamagrostis* host grasses: *B. sylvaticum* (Bs), *B. pinnatum* (Bp), *B. phoenicoides* (Bph), *C. arundinacea* (Ca), *C. purpurea* (Cp) and *C. villosa* (Cv). Asexual isolates are indicated with filled squares. Included are representative sequences of all other sexual *Epichloë* species or subspecies described to date. Dashed lines with Roman numerals denote distinct subclades. The tree is midpoint rooted at the left edge. Branch support values were estimated by 100 ML bootstrap replicates and are indicated if above 50%.

**Table 1 jof-08-01086-t001:** *Epichloë* isolates from *Brachypodium* and *Calamagrostis* hosts used in this study with mode of reproduction (sexual or asexual) and source of isolation (culm, culm with stroma, seed, stroma tissue or ascopores). In addition, use for microsatellite analysis (+) and GenBank accession numbers are indicated (new sequences of this study are in bold).

Isolate No.	Host Plant	Collection Site	Country	Reproduction	Isolation	Micro-Satellites	tubB	tefA
8927/2	*Brachypodium sylvaticum*	Albisgüetli, Zürich	Switzerland	asexual	culm	+	–	–
8928/2	*Brachypodium sylvaticum*	Uetliberg, Zürich	Switzerland	sexual	ascospores	+	KC296736	KC296740
9028/3	*Brachypodium sylvaticum*	Albisgüetli, Zürich	Switzerland	sexual	ascospores	+	**MW283338**	**MW283379**
9127/1	*Brachypodium sylvaticum*	Weissenstein, Solothurn	Switzerland	sexual	culm (stroma)	+	**MW283339**	**MW283380**
9251/2	*Brachypodium sylvaticum*	Sihlwald, Zürich	Switzerland	sexual	ascospores	+	**MW283340**	**MW283381**
9301/1	*Brachypodium sylvaticum*	Sihlwald, Zürich	Switzerland	asexual	seed	+	L78291	AF231219
9344	*Brachypodium sylvaticum*	Uetliberg, Zürich	Switzerland	sexual	culm	+	**MW283341**	**MW283382**
9504	*Brachypodium sylvaticum*	Nishinasuno, Tochigi Pref.	Japan	sexual	culm (stroma)	+	L78278	AF231218
9701 ^a^	*Brachypodium sylvaticum*	Albisgüetli, Zürich	Switzerland	sexual	culm	+	–	–
9702 ^a^	*Brachypodium sylvaticum*	Uetliberg, Zürich	Switzerland	sexual	culm	+	–	–
9703 ^a^	*Brachypodium sylvaticum*	Sihlwald, Zürich	Switzerland	sexual	culm	+	–	–
9704 ^a^	*Brachypodium sylvaticum*	Sihlwald, Zürich	Switzerland	sexual	culm	+	–	–
9705 ^a^	*Brachypodium sylvaticum*	Sihlwald, Zürich	Switzerland	asexual	culm	+	–	–
9806/4	*Brachypodium sylvaticum*	Zollikerberg, Zürich	Switzerland	sexual	ascospores	+	**MW283342**	**MW283383**
9808/1	*Brachypodium sylvaticum*	Parheng, Lund	Sweden	asexual	seed	+	**MW283343**	**MW283384**
9809/1	*Brachypodium sylvaticum*	Bächkrälen, Lund	Sweden	asexual	seed	+	**MW283344**	–
9810/1	*Brachypodium sylvaticum*	Bächriks, Lund	Sweden	asexual	seed	+	**MW283345**	–
9811/1	*Brachypodium sylvaticum*	Stokkem, Zuid-Limburg	Netherlands	asexual	seed	+	**MW283346**	**MW283385**
9812/1	*Brachypodium sylvaticum*	Schiepersberg, Zuid-Limburg	Netherlands	asexual	seed	+	**MW283347**	–
9813/1	*Brachypodium sylvaticum*	Portofino, Liguria	Italy	asexual	seed	+	JF718489	JF718542
9814/1	*Brachypodium sylvaticum*	Tongwynlais, Cardiff	U.K.	asexual	seed	+	**MW283348**	**MW283386**
9815/1	*Brachypodium sylvaticum*	Chéserex, Vaud	Switzerland	sexual	culm (stroma)	+	**MW283349**	**MW283387**
9817/1	*Brachypodium sylvaticum*	Chéserex, Vaud	Switzerland	sexual	culm	+	–	–
9818/2	*Brachypodium sylvaticum*	Kirchspiel Hållnäs, Uppsala	Sweden	asexual	seed	+	**MW283350**	–
9819/1	*Brachypodium sylvaticum*	Kirchspiel Hållnäs, Uppsala	Sweden	asexual	seed	+	**MW283351**	**MW283388**
0731/1	*Brachypodium sylvaticum*	Hinter Guldenen, Zürich	Switzerland	sexual	stroma	+	KC296737	KC296741
1301/1	*Brachypodium sylvaticum*	Valle di Oddeone, Sardinia	Italy	asexual	culm	+	**MW283334**	**MW283375**
1753/2	*Brachypodium sylvaticum*	Pisciotta, Campania	Italy	asexual	culm	+	**MW283335**	**MW283376**
1809/1	*Brachypodium sylvaticum*	Montemayor, Salamanca	Spain	sexual	stroma	+	**MW283336**	**MW283377**
1902	*Brachypodium sylvaticum*	Policoro, Basilicata	Italy	sexual	stroma	+	**MW283337**	**MW283378**
9410	*Brachypodium pinnatum*	Perroudaz, Vaud	Switzerland	sexual	culm	+	L78292	AF231223
9435/1	*Brachypodium pinnatum*	Zumikon, Zürich	Switzerland	sexual	ascospores	+	**MW283331**	**MW283372**
9612	*Brachypodium pinnatum*	Zumikon, Zürich	Switzerland	sexual	culm	+	–	–
9613	*Brachypodium pinnatum*	Zumikon, Zürich	Switzerland	sexual	culm	+	–	–
9639	*Brachypodium pinnatum*	Zumikon, Zürich	Switzerland	sexual	culm	+	–	–
9640/1	*Brachypodium pinnatum*	Zumikon, Zürich	Switzerland	sexual	culm	+	**MW283332**	**MW283373**
0728	*Brachypodium pinnatum*	Eschikon, Zürich	Switzerland	sexual	culm	+	–	–
0901/1	*Brachypodium pinnatum*	Hitzfelden, Alsace	France	sexual	stroma	+	JF718488	JF718541
1407/1	*Brachypodium pinnatum*	Linguaglossa, Sicily	Italy	asexual	culm	+	**MW283328**	**MW283369**
1604/2	*Brachypodium pinnatum*	La Rippe, Vaud	Switzerland	sexual	stroma	+	–	–
1901/1	*Brachypodium pinnatum*	Marsicovetere, Basilicata	Italy	sexual	stroma	+	**MW283329**	**MW283370**
2010/1	*Brachypodium pinnatum*	Ftan, Grisons	Switzerland	sexual	stroma	–	**MW283330**	**MW283371**
2001	*Brachypodium phoenicoides*	Torres del Carrizal, Zamora	Spain	sexual	culm	+	**MW283333**	**MW283374**
1614/1	*Calamagrostis arundinacea*	Ftan, Grisons	Switzerland	sexual	stroma	+	**MW283352**	**MW283389**
1614/2	*Calamagrostis arundinacea*	Ftan, Grisons	Switzerland	sexual	stroma	–	**MW283353**	**MW283390**
2015/1	*Calamagrostis arundinacea*	Ftan, Grisons	Switzerland	sexual	stroma	–	**MW283354**	**MW283391**
0908/1	*Calamagrostis purpurea*	Paltamo, Oulu	Finland	sexual	stroma	–	**MW283355**	**MW283392**
9039	*Calamagrostis villosa*	Lavin, Grisons	Switzerland	sexual	culm (stroma)	–	L78270	AF231196
9618/1	*Calamagrostis villosa*	Bever, Grisons	Switzerland	sexual	culm (stroma)	–	**MW283357**	**MW283394**
0430/1	*Calamagrostis villosa*	Albula, Grisons	Switzerland	sexual	stroma	–	**MW283356**	**MW283393**

^a^ used for mating tests [21].

**Table 2 jof-08-01086-t002:** Microsatellite loci from *Epichloë* spp. with repeat motifs, primer sequences and size ranges used in four fluorescent labelled multiplexes (M1–M4).

Multiplex	Locus	Dye Label	Repeat Motif	Primer Sequence (5′-3′)	Size Range (bp)
M1	E8	6-FAM	(AC)_14_	F: CATGGACCAAGTTGTGAGACC	216–266
				R: AGCAAGTCTCGTAACGGTCTG	
	E39	6-FAM	(GTTTC)_12_	F: GTAGCACATGCATCGAATCAG	408–554
				R: ACCCACTAAAGACGGATGACA	
	E29	VIC	(AGC)_9_	F: TTCCAGCAGCTCTTCAATACC	123–182
				R: ACAGTGGTTCCTGAGGTTTGA	
	E50	VIC	(TTG)_12_	F: TCGTCTTGGACTTTGCCTTT	312–369
				R: TTGAGGTTGTCGAGATACACG	
M2	E13	6-FAM	(GA)_11_	F: GTTCTCCAAGGCTTCCAATTT	464–508
				R: GAGAAACGATATTCGCATTGG	
	E47	VIC	(CTCA)_9_	F: GCCTGTTGAGAAAGACGTGAT	284–320
				R: GATCGAAACACGGGATCATAC	
	E32	NED	(CAG)_11_	F: AGATGAATGGTCAGCAGTTCC	316–343
				R: GGACCATACTTCGTCAACGTC	
	E45	NED	(GT)_15_	F: TTGACGTCGGGAGGTAGTAGA	392–432
				R: CTGGTTACGGAAAGCGAGATA	
M3	E4	6-FAM	(AG)_9_	F: ATTGACCTGTAGCGCGAGTAG	120–134
				R: CAGAACCAATTCGAATCCATC	
	E33	6-FAM	(TCG)_11_	F: TGCCAGATGTTTCAATGACTG	326–334
				R: AACCCATACTCAGCTTTGCAG	
	E36	VIC	(TGC)_7_	F: ATTCGAGAATGGATGACCTGA	402–414
				R: AAGAAAGGAATGGGATTGCTC	
	E41	VIC	(TG)_11_	F: ATTGCCCTGCAGAAGTTGTTA	304–350
				R: TGAGTCGATCGAGAACAAAGA	
M4	E22	6-FAM	(TGGA)_10_	F: GCAAGGATTGGTTGGTGATAA	127–171
				R: GCGGATCACTCTGTAGGCTAA	
	E11	6-FAM	(CT)_11_	F: GTCAGAGGGCAGTAGTGACG	264–280
				R: ATGTAATGCTCTGCCTGCTTC	
	E27	VIC	(GA)_8_	F: TATAAATGACGCTGGGCTTGT	365–391
				R: TGCACTTGAAGAAGCCATGTA	
	E46	NED	(AG)_9_	F: TCGTGACACCTTCTTCGGTAT	376–392
				R: AGAGGTTGTCGTGAGCATCAT	

## Data Availability

DNA sequences are available at GenBank (www.ncbi.nlm.nih.gov).

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
