# Peer review of "Genetic Diversity of *Epichloë* Endophytes Associated with *Brachypodium* and *Calamagrostis* Host Grass Genera including Two New Species"

_jof, 2022, doi:10.3390/jof8101086_

Round 1

Reviewer 1 Report

The manuscript by Leuchtmann and Schardl reports an interesting study of the Epichloe endophytes in Brachypodium and Calamagrostis. They determined the Epichloe sp. infecting a large collection of plants, particularly Brachypodium, from many locations in Europe. The interesting finding is that some of these Epichloe species are not specific as to host Brachypodium or Calamagrostis spp.  This is a well done study and is well written and will be of interest to other researchers interested in Epichloe . I have only a few suggestions.

Table 1: Some of the entries for country and accession numbers are on multiple lines. A smaller font should correct this.

Figure 1: The symbols are not on the version I see.

Figure 3: E. poae (Pn xxx) the accession number is wrong. In the legend should add information as to what the host plant species abbreviations are, in addition to those given for Brachyppodium and Calamagrostis.

The supplementary files were not provided on the website.

According to the journal guidelines the references should be numbered in the text.

Author Response

Table 1: Some of the entries for country and accession numbers are on multiple lines. A smaller font should correct this.

The table is now formated to fit single lines.

Figure 1: The symbols are not on the version I see.

This figure is corrected to show all symbols (this was a problem introdued by journal editors).

Figure 3: E. poae (Pn xxx) the accession number is wrong. In the legend should add information as to what the host plant species abbreviations are, in addition to those given for Brachyppodium and Calamagrostis.

The accession number is now corrected (thanks for noticing this!).

We have decided to omit host abbreviations for all reference strains as this information can be accessed through GenBank and is not relevant for the purpose of this study.

The supplementary files were not provided on the website.

My understanding is that supplementary files will be provided upon publishing of the article.

According to the journal guidelines the references should be numbered in the text.

The refernces are now formated according guidelines.

Reviewer 2 Report

Overall, the manuscript describes comprehensive research work and is of high relevance for phylogeny and evolutionary diversification of Epichloë endophytes of wild Brachypodium and Calamagrostis host grasses.

Author Response

no changes made

Reviewer 3 Report

References missing on M&M when authors mentioning phylogenetic softwares

line 423 to 424 is it a typo ? please fix that

line 456 to 457 is it a legend for a figure? if so it can be aligned properly

are the sequence etc have been submitted to NCBI?

Author Response

References missing on M&M when authors mentioning phylogenetic softwares

We have added Swofford 2003

line 423 to 424 is it a typo ? please fix that

This is fixed now (was an error of the automatic hyphenation)

line 456 to 457 is it a legend for a figure? if so it can be aligned properly

This was a layout problem and is now fixed

are the sequence etc have been submitted to NCBI?

We have added reference to the website (www.ncbi.nlm.nih.gov)